EMBO
Molecular Medicine

# Efficient transduction and optogenetic stimulation of retinal bipolar cells by a synthetic adeno-associated virus capsid and promoter

Therese Cronin[1,2,**], Luk H Vandenberghe[1,3], Péter Hantz[2], Josephine Juttner[2], Andreas Reimann[2], Ágota–Enikő Kacsó[4], Rachel M Huckfeldt[1], Volker Busskamp[2,5], Hubertus Kohler[2], Pamela S Lagali[2,6], Botond Roska[2] & Jean Bennett[1,*]

## Abstract

In this report, we describe the development of a modified adeno-associated virus (AAV) capsid and promoter for transduction of retinal ON-bipolar cells. The bipolar cells, which are post-synaptic to the photoreceptors, are important retinal targets for both basic and preclinical research. In particular, a therapeutic strategy under investigation for advanced forms of blindness involves using optogenetic molecules to render ON-bipolar cells light-sensitive. Currently, delivery of adequate levels of gene expression is a limiting step for this approach. The synthetic AAV capsid and promoter described here achieves high level of optogenetic transgene expression in ON-bipolar cells. This evokes high-frequency (~100 Hz) spiking responses in ganglion cells of previously blind, *rd1*, mice. Our vector is a promising vehicle for further development toward potential clinical use.

**Keywords** adeno-associated virus; capsid library; multi-electrode array; optogenetics; promoter optimization

**Subject Categories** Genetics, Gene Therapy & Genetic Disease; Neuroscience

## Introduction

Much of the early-stage processing of visual inputs occurs within the retinal bipolar cells, and the failure of available vectors to effectively transduce these cells has impacted both basic research and blindness therapies. In this article, we describe a synthetic adeno-associated virus (AAV) capsid and promoter for transduction of ON-center bipolar cells.

The bipolar cells carry the light-induced vertical flow of information from photoreceptors in the outer part of the retina to ganglion cells in the inner part of the retina. Thereby, they serve not only as a conduit from outer to inner retina: The bipolar cells are also "integrating centers" for the retina's response to light (Masland, 2012). To facilitate downstream processing by ganglion cells, the bipolar cells need to combine information from two sources: information coming from photoreceptors on the light hitting the center of the receptor field, and indirect information pertaining to the surround, as shaped by inhibitory inputs from neighboring cells. A second level of processing is achieved by having the two classes of bipolar cells, ON-center (ON-bipolars) and OFF-center (OFF-bipolars), provide parallel information channels responding to increases versus decreases in light intensity. Extensive molecular and physiological studies have revealed how bipolar cells transform information arriving from photoreceptors, and transmit it to retinal ganglion cells (Masland, 2012). *In vitro* systems, transgenic models, and mutation analysis have been used to great effect to characterize the bipolar cells' differing glutamate receptors, signaling proteins, and ion channels (Hanna & Calkins, 2007; Dhingra *et al*, 2008; Audo *et al*, 2009; Peachey *et al*, 2012; Ray *et al*, 2014). However, the molecular effort has been hampered by the lack of a suitable vector to further probe bipolar cell function through conditional labeling and gene knockdown studies.

The need for such a vector is especially pressing in translational research. Recent efforts have pursued a therapeutic strategy using optogenetic molecules to render inner retinal cells light-sensitive (Bi *et al*, 2006; Lagali *et al*, 2008). These efforts have revealed how light-activated channels, such as the various channelrhodopsins

1  Center for Advanced Retinal and Ophthalmic Therapeutics, F.M. Kirby Center for Molecular Ophthalmology, Scheie Eye Institute, University of Pennsylvania, Philadelphia, PA, USA
2  Friedrich Miescher Institute for Biomedical Research, Basel, Switzerland
3  Schepens Eye Research Institute, Massachusetts Eye and Ear Infirmary, Harvard Medical School, Boston, MA, USA
4  Faculty of Physics, Babes-Bolyai University, Cluj-Napoca, Romania
5  Genetics Department, Harvard Medical School, Boston, MA, USA
6  Regenerative Medicine Program, Ottawa Hospital Research Institute, Ottawa, ON, Canada
   *Corresponding author. Tel: +1 215 898 0915; E-mail: jebennet@mail.med.upenn.edu
   **Corresponding author. Tel: +41 61 69 78681; E-mail: therese.cronin@fmi.ch

(Nagel et al, 2003), could be used to restore visual signal in retinas that have lost photoreceptors due to degeneration. Expressed in retinal neurons, the channel will, upon illumination of the cell, bypass the phototransduction cascade and enable direct membrane depolarization. Channelrhodopsin-2 (ChR2) expression in the ON-bipolar cells of retinas from blind *rd1* mice has been shown to make these cells capable of detecting, and also processing, the input visual signal before relaying the information to the ganglion cells and restoring some sensitivity to increasing light (Lagali et al, 2008; Doroudchi et al, 2011). However, efficient targeting of ON-bipolar cells with AAVs expressing sufficient levels of ChR2 to allow ganglion cells to fire at high action potential frequencies (~100 Hz) has not been achieved. This likely requires not only optimized optogenetic tools (Williams & Deisseroth, 2013), but also optimized AAV capsids for ON-bipolar cell targeting and specific promoters elements to improve optogenetic expression.

Vectors based on the small (25 nm) AAV offer the best chance for effective gene delivery. Recent clinical trials using this vector platform have proven to be safe and efficacious in retinal gene therapy (Bainbridge et al, 2008; Hauswirth et al, 2008; Maguire et al, 2008; Simonelli et al, 2010). Follow-up studies have also shown AAV to be effective in its delivery role and in maintaining an excellent safety record for an extended duration of treatment (Maguire et al, 2008; Cideciyan et al, 2009). To circumvent some of its limitations, such as tissue penetration, efforts have been made to reengineer the virus (Muzyczka & Warrington, 2005; Mitchell et al, 2010; Bartel et al, 2011; Yang et al, 2011; Kienle et al, 2012), notably generating an AAV variant capable of reaching the outer nuclear layer from the vitreous (Dalkara et al, 2013). In addition, the retargeting properties of AAV mutants derived through random peptide insertion at the heparan sulfate proteoglycan (HSPG) receptor binding site of the AAV capsid have been investigated (Girod et al, 1999; Perabo et al, 2006). More recently retargeting has been achieved by altering the region of the AAV8 and AAV9 capsid that corresponds to the HSPG-binding site of AAV2 (Michelfelder et al, 2011).

In this work, we have adapted the capsid library approach to enhance AAV-mediated retinal bipolar cell expression. A favorable variant, derived through directed selection from a library of serotype AAV8 capsid mutants, was identified. In parallel, we restricted gene expression to the ON-bipolar cells by further amplifying expression from a validated enhancer sequence of the ON-bipolar-cell-specific gene *Grm6*. We estimate that this synthetic AAV/promoter combination can drive GFP expression in 59% of the ON-bipolar cells in mice. This vector will enable further investigation of the role that bipolar cells play in retinal processing and has the potential to recruit the bipolar cells for retinal gene therapy.

## Results

### Rationale for targeted mutagenesis of AAV8 capsid

Of the many cell types in the retina, the bipolar cells have thus far been the least amenable to transduction by AAV. This may be due to a lack of appropriate receptor expression on the cell surface to mediate uptake, or the presence of an extracellular inhibitory factor on or near the cell surface, or intrinsic factors governing viral trafficking and processing in the cell, or a lack of an appropriate

promoter. We argued that the lack of AAV-mediated expression in the bipolar cells is partly due to failure of the capsid to bind and transduce this cell type. When injected subretinally, AAV2/8 was shown to penetrate deeper into the retinal layers than other serotypes (Vandenberghe et al, 2011). Due to this capacity to "reach" the bipolar cell layer, the AAV8 capsid was chosen as the template for modification.

A 9-amino acid stretch of a conformationally variable region of the AAV8 capsid protein between positions 585 and 593 was specifically selected for its potential to alter receptor attachment and cellular transduction properties of the virus. This was based on information from 3-D models and targeted mutagenesis studies of the capsid (Xie et al, 2002; Padron et al, 2005; Nam et al, 2007; Gurda et al, 2012). The AAV shell is assembled from 60 copies of viral proteins (VP), VP1 (87 kDa), VP2 (73 kDa), and VP3 (61 kDa). The conserved core of each VP subunit consists of an eight-stranded β-barrel motif and an α-helix (Xie et al, 2002). The outer surface of the capsid is formed of large loops that connect the strands of the β-barrel. For example, the residues from amino acids 585–594 of the AAV8 capsid protein encompass finger-like loops for one VP subunit. The amino acid sequences and structural topology of these large outer loops have been reported to facilitate tissue tropism and transduction efficiency (Agbandje-McKenna & Kleinschmidt, 2011). The loop formed by the residues from 585 to 594 contributes to the top of the protrusions that surround the icosahedral threefold axes that are formed through symmetric interactions between the VPs. Thus, this 9-amino acid region holds a prominent position on the capsid and includes sites shown in some serotypes (notably AAV2) to be critical for heparan sulfate binding and cellular uptake (Kern et al, 2003; Opie et al, 2003). The nine residues, which are conserved in many serotypes, are altered in the AAV8 capsid, suggesting that AAV8 does not show any affinity for heparan sulfate (Wobus et al, 2000; Wu et al, 2006). Furthermore, an analogous domain within the heparan sulfate binding region was previously interchanged between AAV serotypes and shown to alter tropism profiles dramatically (Asokan et al, 2010). It is possible that this surface-exposed epitope region of residues 585–594 in AAV8 is amenable to changes that may influence tissue tropism and transduction characteristics and still yield viable capsids.

### Strategy used to modify AAV tropism

From crystallography studies carried out at 2.6-Å resolution, Nam et al have attributed the lack of heparan sulfate binding by AAV8 to be in part due to the structural differences in the region utilized for receptor recognition by AAV2 [highlighted in red in Fig 1A, from PDB 2QA0, (Nam et al, 2007)]. Two critical residues, R585 and R588, are particularly necessary in order for AAV2 to bind heparan sulfate (Kern et al, 2003; Opie et al, 2003). These positions align with Q588 and T591, respectively, in AAV8 (Lochrie et al, 2006). To produce an AAV capsid library, the residues from amino acids 585 to 593 of the AAV8 capsid were replaced with a mixture of sequences to produce randomized codons (Fig 1B). To maximize diversity while reducing the chances of introducing a premature stop, the NNK saturation mutagenesis strategy was applied which should eliminate all stops with the exception of TAG (Muranaka et al, 2006).

This mutated capsid region was cloned into the pAAV8Caps-Lib vector designed for AAV library production and encoding a red fluorescent protein (Supplementary Fig S1B). This plasmid, which has the AAV2 ITRs flanking the minimal rep/cap gene of AAV8 positioned in reverse orientation to the cDNA for dsRed, was used to produce the AAV viral library. An estimated plasmid library degeneracy of $2.6 \times 10^5$ cfu/ml was derived from colony-counting of plated test ligation following initial transformation.

The plasmid was cotransfected with a helper plasmid into HEK293 cells for library production and preparation. An infectious titer of $1 \times 10^5$ IU/µl was estimated for the resulting viral library by limiting dilution analysis on HEK293 cells. Up to 2 µl of the viral library was subretinally injected into Grm6-GFP transgenic mice, in which the ON-bipolar cells are labeled with EGFP expressed under the control of the Grm6 promoter. After 3 weeks, cell dissociation and cell sorting (FACS) were used to isolate the ON-bipolar cells. For most of the tested serotypes, AAV-mediated retinal expression has been shown to require 2–3 weeks to reach optimal levels (Stieger *et al*, 2011). These cells included a subset of cells double-labeled with EGFP and dsRED, that is, library-transfected ON-bipolar cells. The sorted EGFP-labeled cells were lysed, and the mutant region of the capsid was amplified by PCR and recloned back into the pAAV8Caps-Lib plasmid for a second round of viral library production (library R1) and injection. The library R1 thus carried a population of virus particles that, through passive selection, were capable of ON-bipolar cell transduction.

However, it was desirable to select for viruses that could effectively compete with the wild-type virus. Therefore, in the second round, the viral library was spiked with the unmutated virus AAV2/8-dsRed, which in effect served as a competing selective force (Fig 1C). From this round, unmutated and mutant sequences were found in double-positive EGFP/dsRED-labeled cells (transduced ON-bipolar cells) with only unmutated AAV8 capsid sequences isolated from the dsRed-positive, EGFP-negative cells (transduced non-bipolar cells) (Fig 1D). The R1 library carries a heterogeneous mix of capsids; each one producing a unique virus at greatly reduced titer compared to the wild-type co-injected AAV2/8-dsRed. Therefore, it is expected that mutant viruses cannot compete with the wild-type virus in the non-bipolar cells of the retina. However, as the wild-type virus is inefficient at transducing bipolar cells, the R1 library-derived viruses can compete for bipolar cell transduction, and the mutant sequences emerge. From the double-positive red/green cells, the AAVs were isolated and sequenced. Six variants were identified in DNA samples from 44 colonies (Table 1). It is notable that the wild-type capsid sequence was considerably more abundant (30 wild-type sequences versus 14 mutant sequences, Table 1). This was expected, due to the very dilute titer for each individual variant. The WT viral titer is $1 \times 10^{12}$ gc/ml, hence the emergence of any mutant sequence against a saturating WT background is significant.

One variant (variant 5) carried a stop codon and was likely to have been carried through the screen when packaged in another viable capsid. During viral production in the packaging HEK293 cell, the stop-codon-containing sequence may have been taken up by a capsid encoded by another AAV genome within the cell. By using a low plasmid/(packaging-cell) ratio, we hope to minimize the number of alternative AAV genomes within the cell. Nonetheless,

due to the random nature of the transfection, it is not guaranteed that the genome encoding a capsid will be the same as the genome packaged within the encoded capsid and this most likely accounts for the presence of AAV8/BP5. We have confirmed that this serotype is non-viable in luminescence assay (Fig 1E). Its persistence in the screen may be accounted for when you consider that the TAG residue may arise 1.56% of the time (1/64) such that across 9 residues its frequency will be 14%. It served as a negative control for titration.

The variant sequences were processed for further analysis, and four variants were further selected based on the titration data from small-scale (Fig 1E and Supplementary Fig S3) and large-scale preps (Fig 1F) as well as structural predictions of the targeted region anticipated to potentially affect receptor/ligand interactions (Supplementary Fig S2).

## Analysis of AAV2/8BP2 in the retina using a non-cell-specific promoter

AAV vectors were created using each of the selected mutant sequences in place of the wild-type AAV8 capsid gene, and this modified rep/cap plasmid was used to produce AAV(EF1α-EGFP) viruses. Variant 6 yielded low titer and therefore was not used in any further analysis. For this initial analysis, the normal recombinant AAV2/8 virus was injected at a titer of $1 \times 10^{13}$ gc/ml and the variant viruses injected at a titer of $1 \times 10^{12}$ gc/ml. Mouse retinal sections were examined 3 weeks after subretinal or intravitreal injection with the selected AAV variants (Fig 2A and B, Supplementary Fig S4). These qualitative tests using a generic promoter led to the selection of AAV2/8BP2 for further investigation. The cell-type transduction pattern was assessed based on cell soma position in the retina, with the strata of the IPL delineated using a marker for choline acetyl transferase (anti-ChAT). The normal recombinant AAV2/8 shows fluorescence in a diverse range of cell types, with the sparsest fluorescence between the outer (OPL) and inner (IPL) plexiform layers, where the bipolar cell subtypes are found (upper panel of Fig 2A and B). For variant 2, the strongest fluorescent staining was found between the OPL and IPL (lower panel Fig 2A and B), with bipolar and amacrine cells stained. In addition, the photoreceptor cell bodies in the outer nuclear layer (ONL) and their outer segments, as well as cells in the ganglion cell layer (GCL), are stained.

Based on these data, a purified large-scale preparation of AAV2/8BP2 was prepared and *in vitro* and *in vivo* transduction was assayed. This virus was confirmed to transduce HEK293 cells *in vitro* in a similar way to the parental AAV2/8, suggesting that the modified epitope was not having a significant negative impact on the natural tropism (Fig 2C). Adult C57Bl6 mice were subretinally injected with genomic-titer-matched AAVs expressing EGFP under the control of the EF1α promoter and made using either the unmutated AAV8 capsid or the mutant AAV8BP2. After 3 weeks, the retinas were dissociated and processed for FACS analysis. Cell counts (Fig 2D) show 20% transduction of retinas injected with AAV2/8 versus 32% transduction of retinas injected with AAV2/8BP2. However, this difference in transduction of total retinal cells is non-significant. In order to determine what proportion of the transduced cells are ON-bipolar cells, a pool of 80,000 cells was taken from each sorted fraction, and the relative expression of retinal genes in the

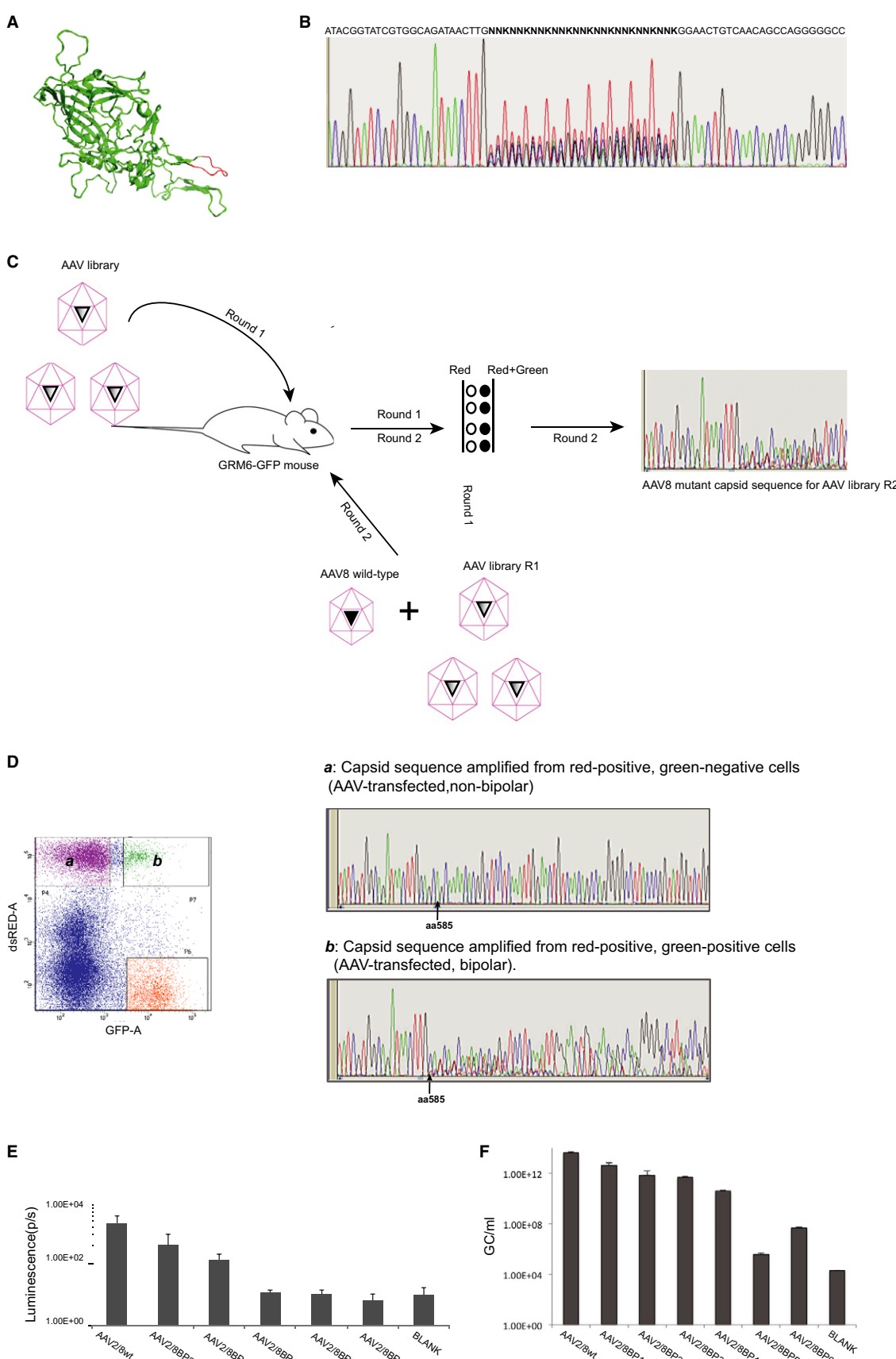

**Figure 1.**

◀

**Figure 1. AAV8 viral library preparation.**

A The secondary structure of the AAV capsid 8 (PDB:2QA0) with the region targeted for mutation highlighted in red.

B Electropherogram of AAV8 capsid sequence from nucleotide 1731 to 1800 containing the mutated region between amino acids 585 and 593.

C The schema for AAV capsid variant isolation: In round 1, the AAV library, "AAV2/8lib-dsRed" was subretinally injected into GRM6-GFP mice. After 4 weeks, the retinas were dissociated and green cells isolated by FACS. In round 2, the viral library filtered through round 1 was used to create a titrated AAV2/8lib-dsRed library R1. This was mixed with a non-mutated AAV2/8-dsRed, serving as competing virus, and the viral mix cosubretinally injected into GRM6-GFP mice. From round 2, green/red and red-only cells were isolated by FACS for further analysis of sequence variants.

D The electropherograms of AAV capsid sequences amplified by PCR from the cells with only non-mutated sequences isolated from red-only cells (a) and non-mutated as well as variant sequences present in the red/green population (b).

E, F The titers of the novel viruses were determined by luciferase assay for small-scale preparations (E) and RT-qPCR for large-scale preparations (F).

green and non-green cell populations was tested using RT-qPCR. The expression of bipolar cell-specific genes, *Grm6* and the long form of a transient receptor potential cation channel, *TrpM1L* (Zeitz *et al*, 2005; Morgans *et al*, 2009), were examined to determine the number of ON-bipolar cells in the transduced cell populations. A 120% increase in *Grm6* expression, and a 67% increase in *TrpM1L* expression, was detected for AAV2/8BP2BP2 compared to AAV2/8-injected retinas, with the gene expression normalized to *β-actin* levels (Fig 2E). In contrast, equivalent expression levels of cone opsin genes were measured between the pools for AAV2/8 versus AAV2/8BP2 retinas (Fig 2F). The gene expression from the cells suggests that improved targeting to the bipolar cells is being achieved with the variant AAV, even when coupled with a strong constitutive promoter such as EF1α. This validated further development of the AAV2/8BP2 virus for bipolar cell-enriched transduction.

## Development of a strong and specific ON-bipolar cell promoter

A specific metabotropic glutamate receptor (mGluR6) is responsible for synaptic transmission from photoreceptors to ON-type bipolar cells (Shiells & Falk, 1994; Masu *et al*, 1995; Ueda *et al*, 1997). This receptor is expressed exclusively in the bipolar cell layer, where it is confined to the postsynaptic site of ON-bipolar cells (Nomura *et al*, 1994; Vardi & Morigiwa, 1997) and is encoded by the *Grm6* gene. The approximate range of the *Grm6* promoter relative to the GRM6 transcriptional start site has been known for some time. Ueda *et al* (1997) generated transgenic mice using the 5′ flanking mouse *Grm6* sequence fused to a reporter gene. This 10-kb region was capable of directing the cell-type-specific and developmentally regulated expression of the *Grm6* gene. Kim *et al* refined the sequence to a 200-bp critical enhancer region (−8126 to −7927 relative to the first

**Table 1. Altered sequence of the AAV8 capsid in amino acid region 585–594 in variants selected from library screen.**
The number of colonies in which the variant was found (out of 44) is listed in the first column.

| | | | | | | | | | |
|---|---|---|---|---|---|---|---|---|---|
| WT 30 colonies | CAG | CAG | CAA | AAC | ACG | GCT | CCT | CAA | ATT |
| | Gln | Gln | Gln | Asn | Thr | Ala | Pro | Gln | Ile |
| | P | P | P | P | (P) | (N) | P[a] | P | (N)[b] |
| Var 1 2 colonies | TAT | CTT | ATG | CGT | TAT | ATT | GGT | GTT | TTT |
| | Tyr | Leu | Met | Arg | Tyr | Ile | Gly | Gly | Phe |
| | P | N[b] | N[b] | B[b] | P | (N)[b] | N | N | N[b] |
| Var 2 4 colonies | CCT | GAG | GGG | ACG | GCG | ATG | AGT | CTT | CCG |
| | Pro | Glu | Arg | Thr | Ala | Met | Ser | Leu | Pro |
| | P[a] | A | B[a] | N | N | N[b] | P | N[b] | P[a] |
| Var 3 3 colonies | AGT | TTT | AGT | CGT | GCG | GTT | CTT | TGT | GAT |
| | Ser | Phe | Ser | Arg | Ala | Val | Leu | Cys | Asp |
| | P | N[b] | P | B[a] | N | N[b] | N[b] | P | A |
| Var 4 2 colonies | CAT | TGT | GTG | GAT | TGT | TGT | GCG | TCT | TAT |
| | His | Cys | Val | Asp | Cys | Cys | Ala | Ser | Tyr |
| | | P | N[b] | | P | P | (N) | P | P |
| Var 5 1 colony | CAT | ACT | GAG | TAT | ATG | AGT | GAG | TAG | CTC |
| | Jis | Thr | Glu | Tyr | Met | Ser | Glu | –[c] | Leu |
| | B[a] | N | A | P | N[b] | P | A | Stop | N[b] |
| Var 6 2 colonies | CCG | ATT | TTT | GTT | GGG | TGT | TCT | GTG | CTT |
| | Pro | Ile | Phe | Val | Gly | Cys | Ser | Val | Leu |
| | P[a] | (N)[b] | N[b] | N[b] | N | P | P | N[b] | N[b] |

B, basic; A, acidic; P, polar; N, non-polar.
[a]Hydrophyllic.
[b]Hydrophobic.
[c]Stop signal.

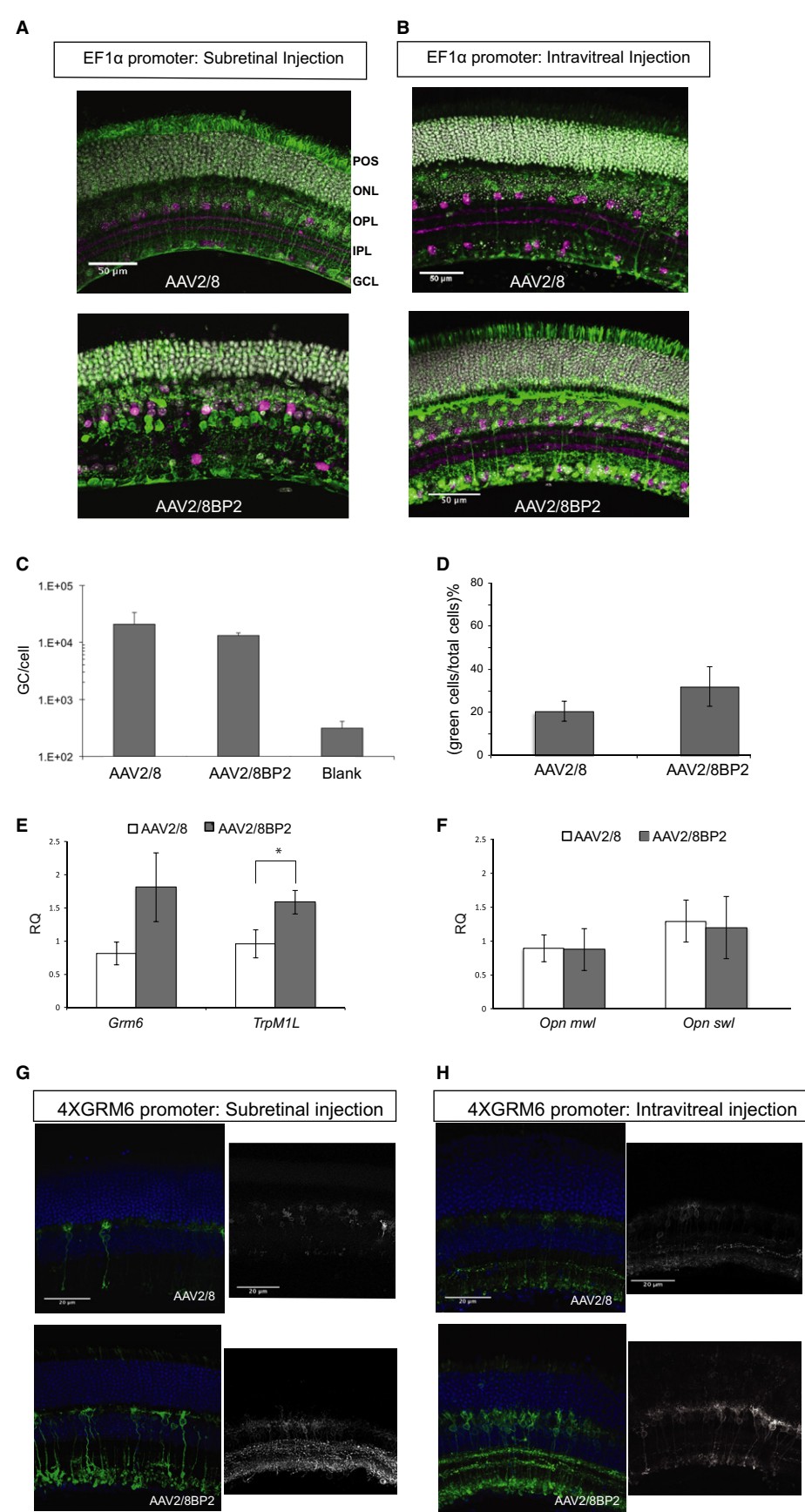

Figure 2.

nucleotide of GenBank accession number BC021919, NIH) that could be used to drive reporter gene expression specifically in ON-bipolar cells (Kim *et al*, 2008; Lagali *et al*, 2008). We tested expression constructs in which the minimal SV40 promoter was preceded by multiple 200-bp enhancer elements. The constructs tested carried 2, 4, 8, or 16 copies of the 200-bp enhancer cassettes in tandem. The strength of reporter gene expression increased with multiple enhancer cassettes, plateauing with the use of four enhancer elements ($4 \times$ Grm6). This arrangement facilitated a high level of expression restricted to the ON-bipolar cells in electroporated mouse retinal explants (Supplementary Fig S5). However, it did not achieve this level of expression when packaged in wild-type AAV vectors and subretinally injected into mice. To test how the variant AAV works in concert with the $4 \times$ Grm6 promoter, we injected C57Bl/6 mice with AAV2/8 or AAV2/8BP2 viruses carrying $4 \times$ Grm6-ChR2-EGFP constructs. Both subretinal and intravitreal injection routes were examined (Fig 2G and H). Both injection routes yield stronger fluorescence using the AAV2/8BP2 virus compared to the AAV2/8 virus. This was evident for stained as well as unfixed and unstained retinas (right-hand panel of Fig 2G and H). Subretinal injection of AAV2/8BP2 gave stronger fluorescence than intravitreal injection, though the latter gave a wider spread of transduction. There was non-specific staining of the ONL and outer segments with the AAV2/8BP2 vector following subretinal injection. At lower titers ($7 \times 10^{12}$ gc/ml), these off-target effects were not observed (Supplementary Fig S6). To facilitate comparison with the wild-type vector, it was necessary to use high titers as no signal was obtained from AAV2/8-$4 \times$ GRM6-injected retinas at titers below $1 \times 10^{12}$ gc/ml.

## Quantification of AAV2/8BP2 transduction in the retina using a bipolar-cell-specific promoter

Images of the retinas at low magnification ($10\times$) show the difference in fluorescence intensity in the INL of retinas injected with the wild-type virus AAV2/8 ($4 \times$ GRM6-EGFP) virus compared to the variant AAV2/8BP2 ($4 \times$ GRM6-EGFP) virus (Fig 3A). The laser power of the confocal microscope was unchanged in comparisons between retinas, and as a result a saturating fluorescence signal was unavoidable in the AAV2/8BP2-$4 \times$ GRM6 retinas. To quantify the improvement in transduction observed, wholemounts of fresh, unfixed retinas from injected mice were used. Local *z*-projections encompassing the ON-bipolar cell body were imaged, and the compressed stack used for fluorescent spot detection (Fig 3B). These cell counts across three randomly chosen regions of each of

five retinas show that an average of 2.3 fluorescent cells were found in a 60 $\mu$m$^2$ area in the AAV2/8 group compared to 89 fluorescent cells for the AAV2/8BP2-injected group (Fig 3C).

In order to test the efficiency of the combined AAV2/8BP2 capsid and $4 \times$ GRM6 promoter in optogenetic stimulations, we created AAVs with AAV2/8BP2-$4 \times$ GRM6 or AAV2/8-$4 \times$ GRM6 capsid–promoter combination, which expressed the channelrhodopsin-2 variant "CatCh". This channelrhodopsin shows reduced desensitization due to a L132C mutation, and as a consequence yields increased photocurrent at a given light intensity (Kleinlogel *et al*, 2011). Before embarking on studies to test light stimulation in the retina, the specificity and efficiency of transduction by these vectors had to be tested. Therefore, immunolabeling for the ON-bipolar-cell-specific protein TrpM1L was carried out on retinas from mice injected with virus expressing $4 \times$ GRM6-CatCh-EGFP (Fig 3F). Counting cells in slices from confocal images of the retinas determined the level of colocalization between the virally expressed GFP and the TrpM1L-immunolabel in cells. The number of colabeled cells was normalized either to the total trpM1L-labeled cells to determine the efficiency of transduction, or to the total number of GFP-labeled cells to determine the specificity of transduction. Compared to wild-type virus, the AAV2/8BP2 retinas showed a threefold increase in ON-bipolar-cell transduction efficiency (Fig 3D). The wild-type and the AAV2/8BP2 retinas showed similar specificity in transduction of bipolar cells, with 80–83% of the green virus-transduced cells also labeling as ON-bipolar cells (Fig 3E).

A FACS analysis was also carried out to more precisely determine the difference in transduced cell number between the variant and wild-type viruses 3 weeks post-injection. We aimed to quantify the range of fluorescence intensity for both viruses and therefore took counts from each of seven subdivisions of the GFP scatter-plot (Fig 3G). Both subretinal (upper panel of Fig 3H) and intravitreal (lower panel of Fig 3H) injection routes were examined. The cells that were brightest, encompassing the subdivisions from Fr12 to Fr14, have a combined percentage of 0.5% of the total retinal cells for mice injected with AAV2/8 versus 4.96% transduction for mice injected with AAV2/8BP2. The RNA from the Fr12-14 pool of cells from AAV2/8BP2 mice was isolated for expression analysis. However, as there were too few corresponding cells in the Fr12-14 cells from AAV2/8 mice this was not included in the analysis. Instead, a comparative study was carried out between this RNA and the RNA from total unsorted retinal cells. We compared the expression of three genes that were shown by Siegert *et al* (2009) to be differentially expressed among the ON-bipolar cell subtypes. *Kcng4*

**Figure 2.  Analysis of transduction properties of AAV2/8BP2 virus.**

A, B   Representative 20× confocal images of immunostained vibratome sections from retinas of mice subretinally injected (A) or intravitreally injected (B) with viruses. The upper panels show AAV2/8(EF1α-EGFP) injected retinas, and the lower panels show AAV2/8BP2(EF1α-EGFP) injected retinas. The retinas are stained for EGFP (green), cell nuclei (gray), and for the inner plexiform layer strata using choline acetyltransferase (ChAT) (magenta). POS, photoreceptor outer segments; ONL, outer nuclear layer; OPL, outer plexiform layer; IPL, inner plexiform layer; GCL, ganglion cell layer.

C   The number of genome copies per cell was estimated for purified AAV2/8BP2 compared to AAV2/8 following transduction of HEK293 cells.

D   Cell counts from FACS analysis of retinas from mice (*n* = 4) subretinally injected with AAV2/8(EF1α-EGFP) or AAV2/8BP2(EF1α-EGFP).

E   RT-qPCR on RNA from the sorted cells used to determine bipolar cell gene expression levels with 120% increase in *Grm6* expression (*P* = 0.05) and 67% increase in *TrpM1L* expression (*P* = 0.04) in the AAV2/8BP2(EF1α-EGFP) cell pool.

F   Equivalent expression levels measured between cell pools for the cone photoreceptor genes *Opnmwl* and *Opnswl*.

G   Representative 40× confocal images of sections from the retinas of mice subretinally injected with AAV2/8($4 \times$ GRM6-EGFP) (upper panel) and AAV2/8BP2 ($4 \times$ GRM6-EGFP) (lower panel). The panel on the left shows sections stained for EGFP (green) and cell nuclei (blue), while the panel on the right shows live fluorescence images.

H   Retinas from mice that were intravitreally injected were similarly analyzed.

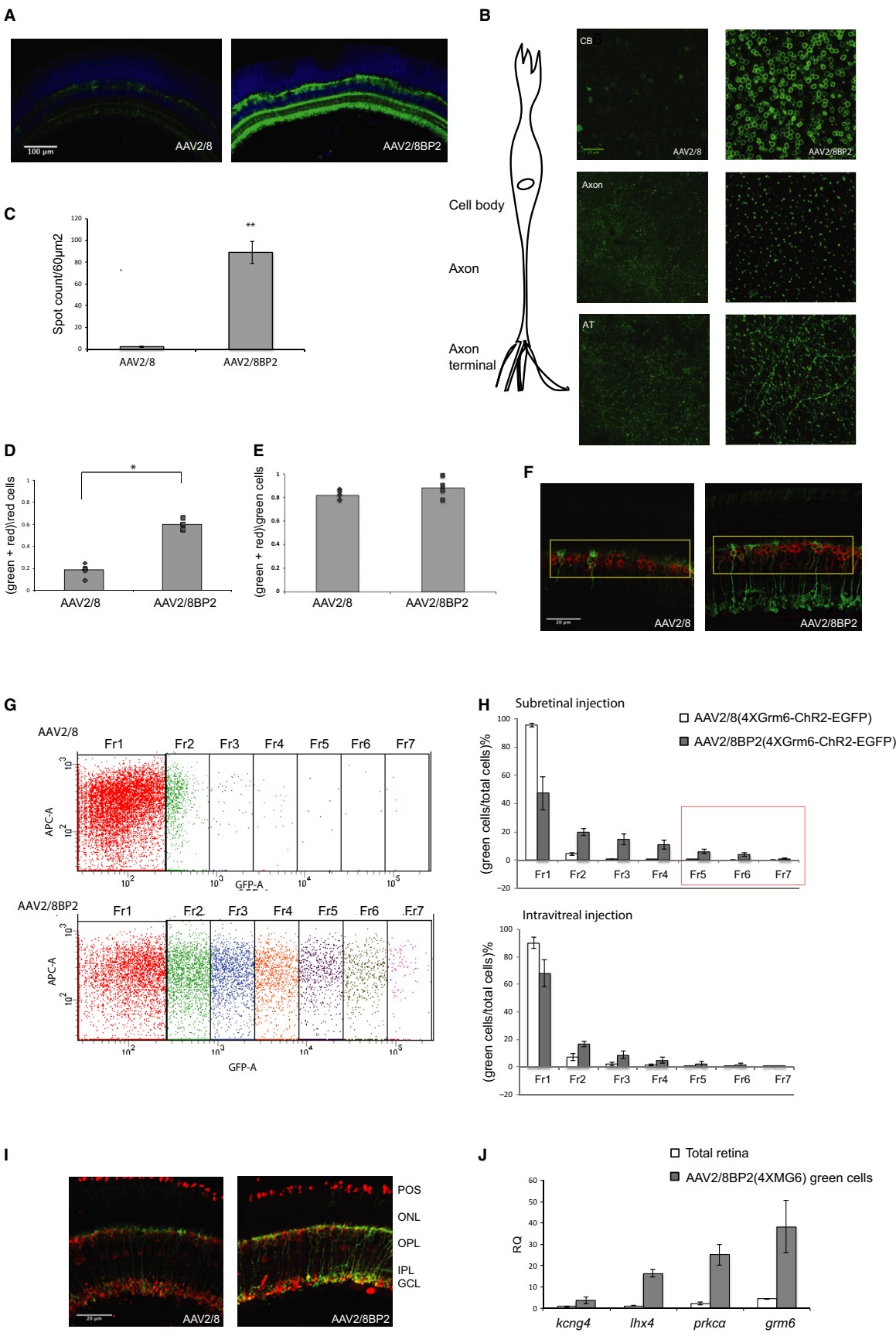

**Figure 3.**

drives expression mostly in type 5, 6, and 7 bipolar cells and showed a fourfold increase in the level of expression in the AAV2/8BP2-4 × GRM6-CatCh-EGFP pool of cells; *lhx4* drives expression in a subset of bipolar cells terminating between chat strata and showed a 13.6-fold increase in expression; the rod-bipolar-cell-specific *prkc-α* showed a 10.4-fold increase, and the pan-ON-bipolar cell marker Grm6 showed a 8.6-fold increase in expression. The high expression of the rod-bipolar-cell-specific protein PKCα in the GFP-pool reflects the high proportion of rod-bipolar cells in the ON-bipolar cell population. Immunostaining for the PKCα protein is shown in retinas of mice subretinally injected with the 4 × GRM6-CatCh-EGFP construct in wild-type AAV2/8 compared to AAV2/8BP2 virus (Fig 3I).

It is also worth noting that no rhodopsin expression was detected from rod photoreceptors, which comprise 70% of the retinal cells, in the RNA from the AAV2/8BP2-4 × GRM6-transduced cell population (Supplementary Fig S7). These results offer us a reasonable estimate of the percentage of ON-bipolar cells we can transduce in the mouse retina. The ON-bipolar cells comprise approximately 7% of the total retinal cells in mice (Jeon *et al*, 1998), and from FACS data, we find 5% transduction of retinal cells following subretinal injection of the AAV2/8BP2-4 × GRM6-CatCh-EGFP virus. TrpM1L-immunolabeling suggests that 83% of these cells are the target ON-bipolar cells (Fig 3E). Therefore, this novel vector is transducing an estimated 59% of ON-bipolar cells. This figure is supported by the TrpM1L-colocalization counts (Fig 3D). However, it is arguably a conservative estimate, and total cell counts on flat-mounted retinas transduced with viruses carrying 4 × GRM6-EGFP suggest the upper limit of transduction (Fig 3B).

## Optogenetic stimulation of AAV2/8BP2-transduced ON-bipolar cells in *rd1* retina

In order to test the efficiency of the combined AAV2/8BP2 capsid and 4 × GRM6 promoter in optogenetic stimulations, we injected *rd1* mice (which lack photoresponses after 1 month of age) subretinally with AAV2/8BP2-4 × GRM6-CatCh-EGFP or AAV2/8-4 × GRM6-CatCh-EGFP and recorded from ganglion cells of mice between 12 and 14 weeks of age using multi-electrode arrays. We were unable to detect any ganglion cell photoresponses in mice injected with AAV2/8-4 × GRM6-CatCh-EGFP except in intrinsically photosensitive retinal ganglion cells (results not shown). However, in mice injected with AAV2/8BP2-4 × GRM6-CatCh-EGFP, ganglion cells responded to light stimulation with short latency, high-frequency spiking (Fig 4). Unlike in previous reports (Lagali *et al*,

2008; Doroudchi *et al*, 2011), we found responses from ON, OFF, and ON-OFF cells (Fig 4A), suggesting that both rod-bipolar and cone ON-bipolar cells were driving ganglion cells. Notably, the average peak firing rate of the recorded ganglion cells reached approximately 100 Hz (Fig 4B and D), and we found several ganglion cells with peak firing in the range of 120–180 Hz (Fig 4B). This contrasts with a previous report using AAV-delivered channelrhodopsin-2 stimulation of ON-bipolar cells where the firing frequency reached 20–25 Hz. As shown before (Lagali *et al*, 2008), we found that stimulation with larger light spots evoked less efficient stimulation than with smaller spots, suggesting the presence of lateral inhibition.

## AAV2/8BP2-mediated transduction of ON-bipolar cells in human retina

In order to determine whether AAV2/8BP2 is capable of transducing human bipolar (and other) retinal cells, the AAV2/8BP2 capsid carrying a cytomegalovirus (CMV) promoted *EGFP* cDNA was used to infect human retinal explants. The CMV promoter was used as the explant system shows downregulation of expression of mGluR6 (data not shown), and CMV has previously been shown to function in this tissue system (Fradot *et al*, 2011). As shown in Supplementary Fig S8, human ON-bipolar cells were GFP-positive and also labeled by immunofluorescence with an antibody against Goα.

## Discussion

It is most likely that early human trials of optogenetics in the coming years will focus on bipolar cells and persisting cones, which have lost light sensitivity. While headway has been made in achieving viral-mediated transduction and restoring function in the cones of blind mouse retinas *in vivo* (Busskamp *et al*, 2010), efficient viral-mediated transduction of optogenetic channels has not been possible in bipolar cells. Nonetheless, Lagali *et al* (2008) demonstrated that some functional properties of image processing could be restored in *rd1* retinas even where only 7% of the bipolar cells express ChR2 delivered by electroporation. Moreover, the ChR2-mediated signals were relayed to the visual cortex and resulted in improvements in visual behavioral tasks. These results hold significant promise for therapeutic outcomes when both the number of bipolar cells expressing the channel and the level of channel expression are increased. Achieving these enhanced levels of expression is the primary goal of this paper. The library screen used to select a virus was designed to

**Figure 3.  Quantitative analysis of transduction properties of AAV8BP2 with a bipolar-cell-specific promoter construct.**

A    10× images of sections from WT mice subretinally injected with AAV2/8(4 × GRM6-EGFP) and AAV2/8BP2(4 × GRM6-EGFP). The sections were stained for EGFP protein expression (green) and for the cell nuclei (blue).

B    Unfixed and unstained whole-mount images of the cell body, axonal and dendritic regions of the bipolar cells from the injected retinas taken at 40× magnification.

C    The fluorescent spot counts on local *z*-stack projections spanning the cell body of these retinas (*n* = 6, *P* = 0.01).

D–F    Cell counts of colocalization of the red (TrpM1L) and green (EGFP) channels (*n* = 4), with the counts normalized to total red cells (D, *P* = 0.04) or to total green cells (E). A representative image of the labeled retinal sections used for cell counting is shown (F).

G    FACS analysis of subretinally and intravitreally injected retinas carried out at 3 weeks post-injection. Representative dot-plot of the fluorescence intensity range for the subretinally injected retinas with GFP versus APC-a (*n* = 6). AAV2/8 injected retina is shown in the top panel while AAV2/8BP2-injected retina is shown in the bottom panel.

H    The FACS intensity range divided across seven regions, Fr8 to Fr14, with the percentage of fluorescent cells shown on the *y*-axis.

I    A representative 40× image of a section from subretinally injected retinas stained for EGFP (green), cell nuclei (blue), and the rod-bipolar cell marker PKCα (red).

J    Expression of *kcng4, lhx4, prkcα,* and *grm6* in cells of sorted fractions Fr12 to Fr14 (highlighted by a red box in the upper panel of Fig 3H) relative to expression in unsorted cells from total retina.

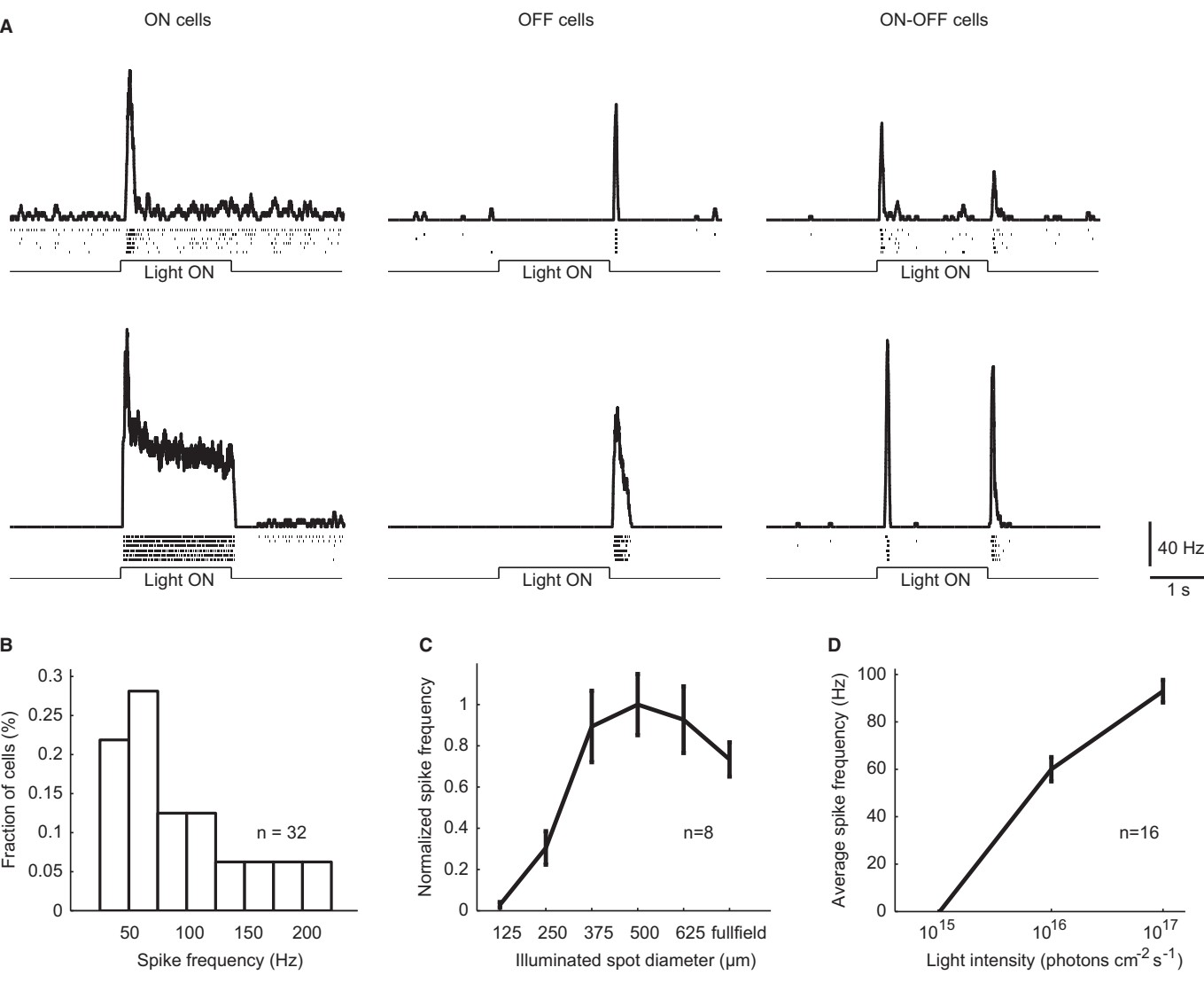

**Figure 4.   Light responses in *rd1* mice after transducing with AAV2/8BP2 expressing CatCh.**

A   Light responses to full field stimulation from six example cells. Both ON, OFF and ON-OFF cells are recorded.
B   Histogram of peak firing frequencies.
C   Response to increasing spots.
D   Firing rate as a function of light intensity.

encompass as much diversity as possible taking into consideration the loss of degeneracy that occurs during library transformation, expansion, and packaging. Many variations on AAV libraries have been done to date, and the success of these libraries relies heavily on the screening process used. We consider the *in vivo* screen using the mglur6-gfp mouse key to the isolation of this virus. Furthermore, the preferential uptake of WT AAV2/8 by non-bipolar cells validated a selection strategy whereby we could isolate variants against a saturating background of WT virus.

The synthetic AAV and promoter combination described here will allow us to further probe the molecular basis of bipolar cell function. However, its immediate promise is a strong candidate for the delivery of light-sensing molecules to the bipolar cell. Bipolar cell-based strategies, which use inner retinal processing, might be used after the loss of cone vision, but before the massive reorganization of the inner retina that probably occurs at later stages of retinal degeneration (Jones *et al*, 2003). Preliminary data from experiments in human retinal explants show GFP-positive ON-bipolar cells after infection with AAV2/8BP2 (Supplementary Fig S8), thereby providing evidence that the viral tropism and transgene expression characteristics in mice and humans are similar. Such transduction of human explants could not be achieved using the unmodified AAV2/8 (Busskamp *et al*, 2010). This argues that the strategy developed in mice can ultimately be extrapolated to humans. Further studies, in progress, aim to evaluate variables that may affect the translational potential. In summary, the work described here is an essential first step in the development of optogenetic agents for the reversal of advanced forms of blindness.

# Materials and Methods

### Terminology

Expression construct for capsid genes isolated from the AAV genome is referred to as pAAV. The plasmid expressing the unmutated AAV8 capsid gene is referred to as pAAV8, and the plasmid expressing the mutated AAV8 capsid gene is referred to as pAAV8BP. The virus carrying the AAV2 Inverted Terminal Repeats (ITRs) and with the unmutated AAV8 capsid is referred to as AAV2/8, while the virus carrying the AAV2 ITRs and with the mutated AAV8 capsid is referred to as AAV2/8BP. For simplicity, the recombinant AAV2/8 virus with unmutated capsid is referred to as the "wild-type" virus.

### AAV library production

The plasmid pRed-Caps-Lib was created incorporating the AAV8 rep/cap gene expressed from a PGK promoter in reverse orientation to the dsRed gene expressed from a CMV promoter. This double cassette was cloned between the AAV2 ITRs. A fragment of the AAV8 rep/cap gene between the sbf1 and mlu1 sites was synthesized by the company DNA2.0 (https://www.dna20.com/). This region includes the amino acids 587–595 (counting from the first amino acid of AAV8 vp1), and mutations were incorporated at each of these nine positions. The plasmid pRed-Caps-Lib was linearized with sbf1 and mlu1. A second sbf1 site further upstream was initially mutated using a site-directed mutagenesis kit (Agilent Technologies, Wilmington, DE), prior to cloning. It was crucial that no single-cut, partially digested backbone from the wild-type plasmid escaped during the cloning as it would contaminate and saturate the library. Initially therefore a stuffer element was cloned into the region of the wild-type capsid between the sbf1 and mlu1 sites to ensure adequate separation of double-cut and partially digested single-cut fragments on a gel. This 3.2-kb stuffer element also carried a kanamycin resistance containing region. Upon restriction enzyme digest of this "pre-library" plasmid (Supplementary Fig S1A) with sbf1 and mlu1, the larger 3.2-kb kanamycin-containing element was excised and replaced with the 370-bp library fragment carrying the 9 NNK residues. As further control, the transformation was screened for no-growth on agar-kanamycin plates. Hence, the final library plasmid contains 4.27 kb between the ITRs as shown in Supplementary Fig S1B. The synthesized mutated region was ligated with the sbf1/mlu1 digested backbone using T4 DNA ligase (New England Biolabs). Three ligation reactions were pooled, ethanol precipitated, and electroporated into DH10b electrocompetent bacteria (Invitrogen).

To titrate the plasmid library, a series of tenfold dilutions from 100 μl of the 1-ml starter culture of transformation broth were plated on agar-ampicillin plates. This sample was taken from the starter culture after 30 min of incubation of the newly transformed bacteria. We make the assumption that each colony is derived from a single random member of the library and that minimal divisions have taken place prior to plating. Therefore, colony-counting can give an estimate of library diversity and a plasmid library diversity of $2.6 \times 10^5$ cfu/ml was counted. The remainder of the starter culture was used to inoculate 2 l of broth to prepare rep/cap plasmid library DNA for AAV production. (Note: To maximize diversity of the plasmid library, the NNK mutagenesis saturation strategy was applied which theoretically covers all 20 amino acids at each of the 9 sites yielding a potential degeneracy at the amino acid level of 5.12E9. It is clear that only a fraction of the combinations theoretically generated was represented in the recovered library and the efficiency could be improved with further optimization of the cloning process). The AAV library was prepared by cotransfection using the plasmid library DNA and a helper plasmid. This was carried out according to standard procedure with some alterations: To minimize the transfection of multiple different rep/cap genes into each cell (which would lead to encapsidation of a non-"self-coding" genome), a low plasmid concentration was used for the triple transfection (Maheshri *et al*, 2006)—150 ng of rep/cap library plasmid was used per 150-mm confluent HEK293 cell culture plate. Ultimately, the transfected cell lysates, and broths from 70 confluent 150-mm transfected plates were used to prepare the library. The broth and benzonase-treated lysates were concentrated 20-fold using a tangential flow filtration system before virus purification using a discontinuous iodixanol gradient (Sigma, Optiprep). To estimate the infectious titer of the AAV library, HEK293 cells were seeded at 50% confluency on poly-L-lysine-coated glass coverslips across 10 wells of a 24-well plate. Serial dilutions of the AAV library were added directly onto each well. After 72 h, the cells were fixed with 4% paraformaldehyde (PFA) and washed with PBS and dH₂O. The number of fluorescent cells from the well with highest dilution factor that still showed infection was recorded and multiplied by the dilution factor to give the number of infectious units per microliter (IU/μl).

### AAV preparation

Standard AAV production of individual viruses was performed as previously described (Grieger *et al*, 2006) by triple transfection of HEK293 cells with branched polyethylenimine (PEI) (Polysciences, no. 23966) with a plasmid containing the transgene between the ITRs of AAV2, the AAV-helper plasmid encoding Rep2 and Cap for serotype variants, and the pHGTI-Adeno1 plasmid harboring helper adenoviral genes (both kindly provided by C. Cepko, Harvard Medical School, Boston, MA, USA). The HEK293 cells express the helper E1A/E1b gene (American Type Culture Collection, catalogue number CRL-157). Vectors were purified using a discontinuous iodixanol gradient (Sigma, Optiprep). Encapsidated DNA was quantified by TaqMan RT–PCR (using primers "AAV titer", Supplementary Table S1), following denaturation of the AAV particles by proteinase-K, and titers were calculated as genome copies (gc) per ml.

### *In vitro* titer assay for individual AAV capsid variants

Capsid genes were cloned in an AAV packaging plasmid for vector production, and used for small-scale vector preparations encoding firefly luciferase to obtain the titer shown (Fig 1E). Physical particle titers were established by TaqMan qPCR (Supplementary Fig S3). Subsequently, AAV2/8BP variants were assayed for transduction at equal multiplicity of infection onto HEK293 cells. For large-scale viral titer, the encapsidated DNA was quantified by TaqMan RT–PCR (using primers "AAV titer", Supplementary Table S1),

following denaturation of the AAV particles by proteinase-K, and titers were calculated as genome copies (gc) per ml (Fig 1F).

## Mice

The GRM6-EGFP mice were kindly provided by Dr. Noga Vardi (University of Pennsylvania). C57Bl/6J were obtained from breeding stock at the Jackson Laboratory (Bar Harbor, ME), and C57Bl/6NCrl and C3HeN mice from breeding stock at the Charles River Laboratories (L'Arbresle Cedex, France). All animal experiments and procedures performed in the US were reviewed and approved according to the guidelines set out in the National Institutes of Health's Guide for Care and Use of Laboratory Animals with approval by the University of Pennsylvania Institute for Animal Care and Use Committee (IACUC), protocols 804546,804543. All animal experiments and procedures performed in Switzerland were approved by the Swiss Veterinary Office. For these experiments, both eyes of the mice were injected using a stereotactic setup. The same researcher (JJ) carried out all the injections to compare AAV2/8BP2 with the normal recombinant virus. Up to 2 μl of virus, at titers ranging from $1 \times 10^{10}$ gc/ml to $1 \times 10^{13}$ gc/ml were injected into both eyes. The numbers of mice used are outlined in Table 2.

## Construction of the 4 × Grm6 promoter

The 4 × Grm6 promoter construct was created by iterative cloning of the single Grm6 enhancer element contained within the pGrm6-CY expression plasmid (Lagali et al, 2008). First, the Grm6 enhancer element was amplified by PCR using the primers "Grm6 enhancer" (Supplementary Table S1) to generate XbaI and SspI restriction sites upstream of the Grm6 sequence, and a SalI site at the 3′ terminus of the enhancer element. The PCR product was blunt-ended with the Klenow fragment of DNA polymerase I and then digested with SalI. This DNA fragment was subsequently inserted into the SspI/SalI double-digested pGrm6-CY plasmid to create two copies of the Grm6 enhancer element in tandem. The resulting plasmid was digested with XbaI and SspI, and the PCR product generated above was inserted via the XbaI-cut 5′ end and Klenow-blunted 3′ end. The previous subcloning step was repeated to insert an additional copy of the enhancer element to yield the 4 × Grm6 promoter construct.

## Immunohistochemistry and imaging

Retinas were dissected from the eyecup, fixed in 4% (wt/vol) paraformaldehyde in PBS for 20–30 min, and washed overnight in PBS. They were incubated in 30% sucrose for 30 min at 22–23°C before being submitted to three freeze–thaw cycles. Wholemounts or 150-μm vertical sections cut with a Leica VT1000S vibratome were used. The retinas were then incubated in blocking solution (10% normal donkey serum (vol/vol, Chemicon), 1% bovine serum albumin (wt/vol), and 0.5% Triton X-100 (vol/vol) in PBS, pH 7.4) for 1 h. Primary and secondary antibody applications were done in 3% normal donkey serum, 1% bovine serum albumin, 0.02% sodium acid (wt/vol), and 0.5% Triton X-100 in PBS. Primary antibodies [rabbit antibody to GFP (1:200; Molecular Probes) and goat antibody to ChAT (1:300; Chemicon)] were applied for 3–7 days. In addition, an antibody against the long isoform of TrpM1 (NP_001034193.2) was raised and purified by Eurogentec. We designed this antibody to target a TrpM1L-isoform-specific epitope (C+PQISRSALTVSDRPE) and used it for ON-bipolar cell labeling. Secondary antibodies were purchased from Invitrogen (Alexa Fluor 488, Alexa Fluor 555, Alexa Fluor 633) or from Jackson Laboratory (Cy3, Cy5) and used at a concentration of 1:200. The wholemounts and retinal sections were mounted on slides with ProLong Gold antifade reagent (Molecular Probes). Confocal three-dimensional scans of 1024 × 1024 pixel images in a z-stack were taken with a Zeiss LSM 700 confocal microscope using three excitation laser lines (405 nm for DAPI, 488 nm for GFP, and 633 nm for ChAT) and imaged using Zen imaging software (Carl Zeiss MicroImaging GmbH). Images were processed using Fiji (http://fiji.sc/wiki/index.php/ImageJA). Equivalent processing was carried out for both test and control 32-bit images. A z-projection was made on fixed stack size, and the gamma changed to visualize low-value pixels. The images were saved as RGB color and imported into Adobe Illustrator. For wholemount retinas, a local stack was made spanning the bipolar cell body in 40× images of the retinas, and the spot detection performed using fixed threshold in Fiji. For the colocalization study, slices from a stack of 40× images acquired for the red and green color channels were processed as composite in Fiji. The "cell counter" plug-in was used to record cells in different color channels using fixed brightness and contrast settings.

**Table 2.  The numbers of mice used in these experiments.**

| Number of mice | Strain | Age | Procedure | Relevant figure |
|---|---|---|---|---|
| 10 | Grm6-EGFP | Injected at 10 weeks, tested at 13 weeks | Subretinal injection of AAVlib | 1A, B, C, D |
| 8 | C57Bl6 | Injected at 5 weeks, tested at 10 weeks | Subretinal injection of AAV variants | 2A, S4 |
| 6 | C57Bl6 | Injected at 6 weeks, tested at 10 weeks | Intravitreal injection of AAV variants | 2B, S4 |
| 8 | C57Bl6 | Injected at 8 weeks, tested at 12 weeks | Subretinal injection to quantitatively test AAV2/8BP2(EF1α-EGFP). | 2D, E, F |
| 15 | C57Bl6 | Injected at 8 weeks, tested at 12 weeks | Subretinal injection to test AAV2/8BP2(4 × Grm6-EGFP) | 2G, 3A, D, E, F, G, H, I, J |
| 6 | C57Bl6 | Injected at 8 weeks, tested at 12 weeks | Intravitreal injection to test AAV2/8BP2(4 × Grm6-EGFP) | 2H, 3B, C, H |
| 5 | C3Hen | Injected at 10 weeks, tested at 14 weeks. | Subretinal injection to test AAV2/8BP2(4 × Grm6-CHr2d-EGFP) responses | 4A, B, C, D |

## FACS analysis and RT-qPCR

Retinas were dissociated as previously described (Siegert *et al*, 2012). The retinas from left and right eyes were processed separately and counted as separate experiments. Cells were sorted immediately following dissociation on a BD FACSAria cell sorter (Becton Dickinson), and 80,000 cells were collected directly in 600 µl of Trizol LS (Life Technologies) for RNA purification according to the manufacturer's protocol. The RNA was quantified using a NanoDrop, and all samples normalized to 35 ng/µl concentration before reverse transcription using random hexamer primers (Roche) and transcriptor reverse transcriptase (Roche). The cDNA was diluted 1/5 and a 3 µl volume used in the qPCR. For the TaqMan assay, the probes outlined in Supplementary Table S2 were used (Actb, *Rho, Grm6, TrpM1L, Opn swl*, and *Opn mwl*) with universal master mix (Applied Biosystems). For SYBR Green assays, the desalted primers outlined in Supplementary Table S1 were used (*Prkcα, Kcng4, Lhx4, Actb*) with SYBR Green reagent (Invitrogen). The mouse *β-actin* gene was used to normalize the expression levels in the TaqMan reaction, and *18srRNA* used in the SYBR Green reaction. Samples were loaded in triplicate where possible. Relative quantification was performed as previously described (Pfaffl, 2001).

## Human retinal explants

Use of human tissue was in compliance with local and federal regulations. De-identified tissue was provided by Miracles in Sight, North Caroline Eye Bank, Winston-Salem, NC, USA, an organization which provides ocular tissue to assist researchers to ultimately find cures for eye conditions/diseases. Experiments conformed to the principles set out in the WMA Declaration of Helsinki and the NIH Belmont Report. A postmortem human retina was obtained and treated with AAV2/8BP2-CMV-GFP at a titer of $4.38 \times 10^{12}$ gc/ml (and delivering 5.5E10 gc) using methods described (Fradot *et al*, 2011). Immunofluorescence was carried out 12 days after infection using a mouse primary antibody against Goα to label ON-bipolar cells (Chemicon; mAB3073; 1:500), an Alexa fluorophore-conjugated secondary antibody (Invitrogen, Grand Island, NY, USA), DAPI, and a FV1000 confocal microscope (Olympus, Center Valley, PA, USA).

## Electrophysiology

The retina was isolated under dim red light in Ringer's medium (in mM: 110 NaCl, 2.5 KCl, 1 $CaCl_2$, 1.6 $MgCl_2$, 10 D-glucose, 22 $NaHCO_3$) bubbled with 5% $CO_2$/95% $O_2$. The retina, ganglion side down, was then immobilized on the multi-electrode array by gentle pressing with a cell culture membrane (Transwell 3450-Clear) having hexagonally arranged holes with 200 µm diameter and a center-to-center distance of 400 µm. For the duration of the experiment, the retina was perfused with Ringer's medium bubbled with 5% $CO_2$/95% $O_2$, at a flow rate of 1.5 ml/min at 35°C. Extracellular voltage was measured with a multi-electrode array (MEA1060 Up-BC amplifier, Multichannel Systems) at 20 kHz. The array was fixed on a motorized table (Scientifica). Light stimulus was generated using a DLP projector (PLUS U137SF) and projected onto the retina by the condenser lens of an inverted microscope (Nikon TE300).

**The paper explained**

**Problem**
Excellent safety and efficacy data relating to AAV-mediated retinal gene augmentation therapy have been collected in early- and late-phase clinical trials in children and adults with a rare, early onset blindness called Leber's Congenital Amaurosis. While this approach could theoretically be used to reverse blindness in other inherited diseases, there are practical limits to its utility, including requirements for intervention before the target cells (photoreceptors/retinal pigment epithelium) have died and identity of the disease-causing gene. There is a large unmet need for a therapy that could be used in any inherited or acquired form of blindness. One possibility for such a broad-based therapy lies in the burgeoning field of optogenetics. In advanced stages of disease where photoreceptors are no longer viable, the inner retinal and CNS visual pathway circuitry can be harnessed by rendering the second-order retinal neurons light-sensitive through delivery of an appropriate optogenetic channel. Until now, this was not possible due to a lack of vectors, which can efficiently and stably transduce such neurons, bipolar cells.

**Results**
The AAV2/8 recombinant virus was used as a template for library screening of viral capsid mutants that would efficiently transduce bipolar cells. In parallel, a bipolar-cell-specific promoter was optimized for strong, cell-specific expression. One capsid variant, together with a transgene cassette carrying the enhanced promoter, showed high and stable levels of transduction of bipolar cells. This vehicle was used to deliver a channelrhodopsin optogenetic molecule to the retinas of the *rd1* mouse. The retina of this mouse typically shows no response to visual stimuli; however, when transduced with the novel channelrhodopsin-expressing virus, robust spiking responses were recorded.

**Impact**
The engineering of this vector represents an essential step in the development of optogenetic agents for the reversal of advanced forms of blindness.

The spectrum of the stimulus light was determined using a spectrophotometer (Ocean Optics USB2000), and the light intensity was measured using a power meter (Thorlabs S130VC). The stimulation intensity was calculated by integrating the product of the projector spectrum and the normalized absorption spectrum of CatCh. The recorded voltage was bandpass-filtered (400–4,000 Hz), and spikes were sorted using the UltraMegaSort software (Kleinfeld Lab, University of California, San Diego). Spike frequency was calculated using 50 ms moving bins. Intrinsically photosensitive retinal ganglion cells were discriminated by their delayed spiking. For quantification, we only used spike frequency values in the first 200 ms after light onset or offset.

## Statistics and software

CLC Main Workbench was used for sequence annotation and *in silico* cloning (CLC-bi, Denmark). PyMOL was used for 3-D visualization of the mutated capsid structures (The PyMOL Molecular Graphics System, Version 1.5.0.4, Schrödinger, LLC.). Zen imaging software (Carl Zeiss MicroImaging GmbH) was used to load images taken by confocal microscopy, and these images were processed using Fiji (http://fiji.sc/wiki/index.php/ImageJA). StepOne software v2 (Applied Biosystems, Life Technologies, Switzerland) was

used to process qPCR data. Excel (Microsoft Office) and GraphPad Prism 6 (GraphPad Software, Inc., CA, USA) were used for all other statistical calculations. The Mann–Whitney $U$-test was used to determine significance in differences between pairs. Data were expressed as the mean $\pm$ standard error of mean (SEM) where $n \geq 7$, and mean $\pm$ standard deviation (SD) where $n < 7$. In the figures, different levels of significance are indicated by * for $P \leq 0.05$, **$P < 0.01$.

**Supplementary information** for this article is available online:
http://embomolmed.embopress.org

## Acknowledgements

The GRM6-GFP mice used for library screening were kindly provided by Noga Vardi (Perelman School of Medicine, University of Pennsylvania). TC was funded by grants from Hope for Vision and the Marie Heim-Vögtlin (MHV) program of the Swiss National Science Foundation. Additional support was provided by NEI/NIH 8DP1 EY023177, 1R24EY019861-01A1, Foundation Fighting Blindness, Research to prevent blindness, FM Kirby Foundation.

## Author contributions

AAV viral libraries were prepared by TC and LV. The original pAAV-Lib was designed by LV. AAVs for comparative viral testing were prepared by JJ. Mouse injections for comparative viral testing were carried out by JJ, and electroporation was carried out by VB. Cell sorting was performed by HK. The 4 × Grm6 promoter was conceived and designed by PL. It was constructed and tested by AR. Histology, RT-qPCR, and data analysis were done by TC. PH and VB performed the physiology experiments, and PH and AK analyzed the MEA experiments. RH carried out and analyzed the human retinal explant experiments. Experiment planning involved TC, LV, BR, and JB, and manuscript preparation was carried out by TC, BR, and JB.

## Conflict of interest

TC, LH, and JB are co-authors on a US patent application for "Enhanced AAV-mediated gene transfer for retinal therapies". The other authors declare that they have no conflict of interest.

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

    Mutations in GRM6 cause autosomal recessive congenital
    stationary night blindness with a distinctive scotopic 15-Hz
    flicker electroretinogram. *Invest Ophthalmol Vis Sci* 46:
    4328−4335

