## [Review Process File · EMBO Molecular Medicine]

Efficient transduction and optogenetic stimulation of retinal bipolar cells by a synthetic adeno-associated virus capsid and promoter

Therese Cronin, Luk H.Vandenberghe, Péter Hantz, Josephine Juttner, Andreas Reimann, Ágota-Enikő Kacsó, Rachel M. Huckfeldt, Volker Busskamp, Hubertus Kohler, Pamela S. Lagali, Botond Roska and Jean Bennett

Corresponding authors: Jean Bennett, Scheie Eye Institute, University of Pennsylvania and Therese Cronin, Friedrich Miescher Institute for Biomedical Research

Review timeline:

Submission date:	17 March 2014
Editorial Decision:	23 April 2014
Revision received:	04 June 2014
Editorial Decision:	23 June 2014
Revision received:	04 July 2014
Accepted:	07 July 2014

Transaction Report:

Editor: Céline Carret

1st Editorial Decision

23 April 2014

Thank you for the submission of your manuscript to EMBO Molecular Medicine. I am sorry that it has taken so long to get back to you on your manuscript.

While reviewers 1 and 2 delivered their evaluations in a timely manner, we did not receive the other reviewers' input. As the evaluations from the first two reviewers are consistent, and a further delay cannot be justified, I have decided to proceed based on these evaluations.

If in the meanwhile we should receive the other review within the next 10 days and only if they raise significant caveats, these would need to be taken into consideration. We would not, however, ask you to comply with any further-reaching requests.

As for the reports below, you will see that while both reviewers are generally supportive of your work and underline its considerable potential interest, they also raise a number of specific concerns that need to be addressed in the next version of your manuscript. Referee 1 suggests modifying the text to make it more suitable to a general audience. EMBO Molecular Medicine does indeed publish studies on various topics that are of general interest. This referee also has minor issues that are self-explanatory. Referee 2 however is a little more critical and would like to see better explanations and more details on the protocol used.

Overall, given these evaluations, I would like to give you the opportunity to revise your manuscript, with the understanding that the referee concerns must be fully addressed and that acceptance of the manuscript may entail a second round of review.

I look forward to receiving your revised manuscript.

***** Reviewer's comments *****

Referee #1 (Comments on Novelty/Model System):

The technical quality of the reported work appears to be very high. As far as I can discern appropriate statistical and quantitative analyses have been carried out to validate the results. The approach is not completely novel since the aim and achievement is to improve a more or less pre-existing approach. The work has been carried out in a retinal dystrophy (rd1) mouse model system. The vector delivery system is not necessarily fully optimised for human tissue-specific delivery and so the medical relevance is judged as medium. The mouse model is adequate as a first step and for the proof of concept. The resulting improved AAV vector may work well in humans as well, but it has only been tried in the human. The mouse is a nocturnal animal with a different retinal structure from that found in humans. However, conceptually the approach used here is important and potentially transferrable to humans.

Referee #1 (Remarks):

This paper addresses the important problem of improving the efficiency of delivering and expressing at sufficiently high levels the optogenetic receptor channelrhodopsin in bipolar cells of the inner retina, to permit direct light stimulation when the usually connecting outer retinal photoreceptors have been rendered non-functional by degeneration. The major task performed was to improve the delivery capacity of the AAV8 virus subtype to bipolar cells specifically, having shown that it seems to penetrate more deeply into the retina. Careful analysis of the structure of AAV viral capsid protein pinpointed a nine aminoacid stretch to target by random mutagenesis and incorporation of the altered region into a hybrid virus followed by selection of improved uptake into bipolar cells, via testing in HEK293 cells. Getting capsid changes to permit more efficient delivery to the relevant target cell is a first step, but high enough expression of the delivered gene is also required. dsRED was used as the reporter for testing the specific delivery into GFP-marked ON-bipolar cells. An improved bipolar cell-specific enhancer/promoter (please clarify nature of element used) was developed from known elements derived from the 5' upstream region of the bipolar cell-specific metabotropic glutamate receptor (mGluR6). A previously identified minimum element of 200 bp was tested and found to function maximally when four copies were used to direct expression of the reporter. The specificity and functionality of the newly developed vector was carefully tested and demonstrated in vivo in wild type mouse retinas and then tested for optogenetic responsiveness in rd1 mutant mice at 12-14 weeks of age. Appropriate controls were used for each stage

Comments on the paper

The work reported is carefully thought-out and elegantly executed.

The results contribute towards proof of principles: the requirement for more efficient cell-specific vector delivery and higher expression levels in the transduced cells. The potential problems of transferring strategy developed in the mouse to humans should be addressed briefly in the discussion.

The paper is not very easy to read, but not because it is not clear linguistically, but because it is written for the cognoscenti.

There are a number of un-referenced statements (eg "In vitro systems, transgenic models, and mutation analysis have been used to great effect to characterize the bipolar cells' differing glutamate receptors, signaling proteins, and ion channels. However, the molecular effort has been hampered by the lack of a suitable vector to further probe bipolar cell function through conditional labeling and gene knockdown studies.").

The nature and identity of HEK293 cells is not referenced. Is their use important for AAV production because they are adeno-transformed? If so please document this. There are several

undefined abbreviations (eg ITR). Please check the clarity of the manuscript further - perhaps by asking a young undergraduate to read it.

Referee #2 (Remarks):

The manuscript Efficient transduction and optogenetic stimulation of retinal bipolar cells by a synthetic adeno-associated virus capsid and promoter by Cronin and colleagues describes the development, testing and optimization of an AAV8 capsid mutant selected from an AAV8 library after two rounds of in vivo selection. The novel capsid was combined with an On-bipolar cell-specific promoter and demonstrated efficient transduction of retinal ON-bipolar cells in vivo. Delivery of channel rhodopsin led to robust spiking responses.

The authors developed a novel library for their screen. To my knowledge this region has been used so far for insertion of random peptides (which should be mentioned in the introduction) or - as described by the authors - for domain swapping. Here, nine residues of the natural loop region were mutagenized. Since a new library was developed and an unusual selection strategy was applied, the authors should provide more information on both. They should determine and compare the diversity of the plasmid and viral library. This will allow drawing conclusions on amino acid residues that are not tolerated and will allow to judge on the quality of the viral library. Since one candidate variant that had been selected (definition of the authors: selected more than once) showed a stop codon mutation, the question arises how the authors ensured that mutations presented at the capsid are encoded by the viral genome delivered by the respective mutant. In addition, how many particles within the initial library showed a wild-type sequence? What is the size of the viral genome packaged into the viral library? From the plasmid map it seems that the viral genome is oversized. What is the diversity of the library and which titer could be reached?

Furthermore, the authors hypothesized that addition of wild-type AAV8 served as a negative competing force. However, in order to make this statement, selection should be performed in parallel in the presence and absence of wild-type AAV8. Since multiple virions can infect one cell, why do the authors speculate that mutant and wild type AAV will compete with each other for cell transduction?

If ON-bipolar cells are poor target cells for AAV8 as stated on page 6, why is the wild type sequence the most prominent sequences that was selected (Table 2).

How is it possible that BP5 could be packaged and showed infectivity although capsid production is impossible through the stop codon mutation (Figure 1E and F)? On the same line, how was it possible to select this sequence?

From the text it is not clear whether comparable particle numbers were injected when comparing the 4 different mutants (Figure 2 and S3).

Did the author observe any cell transduction outside the retina? In other words, did the authors perform a bio-distribution analysis?

The authors state that they used a low plasmid concentration to avoid packaging of multiple genomes into each capsid. It is not possible to package multiple full-length genomes into a single AAV capsid. Thus, this statement needs to be corrected.

Minor points

The author should add that information of 3D models and targeted mutagenesis studies included AAV8, but also further serotypes (page 4).

Yellow box in Figure 3 is a red box

Label of Figure S6 should be improved

To the reviewers:

We are grateful to the reviewers for taking the time to thoroughly evaluate our article. From their insights we note that there are two overriding concerns that need to be addressed: 1) the therapeutic context of the work should be highlighted; 2) more clarity is required on the process used to produce the viral library and on the interpretation of the results derived from the library.

Please find below point-by-point responses to the comments of the reviewers.

Reviewer #1:

1) Therapeutic context

“The potential problems of transferring strategy developed in mouse to humans should be addressed”

The problems that might arise in the transfer of the viral vector from mouse to human application are addressed in in the Discussion, page 16. Preliminary data suggest that the strategy developed in mice can be extrapolated to humans, as the recombinant virus has a similar transduction profile in human retinal explants as in retinas of mice. This is the first and critical step in determining whether there are any species differences in transduction characteristics and optimizing expression for applications to the primate retina.

Reviewer #2:

2) Viral library

“To my knowledge this region has been used so far for insertion of random peptides (which should be mentioned in the introduction)”

In the Introduction, we now refer to two articles describing work that targeted this region, either to investigate receptor function or to create a library (page 3).

“They should determine and compare the diversity of the plasmid and viral library”

We have added the following to the results section (page 5):

“An estimated plasmid library degeneracy of 2.6×10^5 cfu/ml was derived from colony counting of plated test ligation following initial transformation”.

And to the methods section we have added:

“To titrate the plasmid library a series of 10-fold dilutions from 100 μ l of the 1ml starter culture of transformation broth were plated on agar-Ampicillin plates. We can assume that each colony is derived from a single random member of the library, and the library diversity was based on a colony-count of 2.6×10^5 cfu/ml”.

In theory a diversity of 5.12×10^9 clones should be possible from the random mutagenesis of each of 9 residues. Moreover to establish a 99% probability that each possible combination is represented at least once in our final plasmid library we would need to have approximately 1.1×10^{15} clones ($n \times \ln(0.01)$ where $n=5.12 \times 10^9$). It is not possible to reach the potential diversity of the plasmid library in our screen. The diversity is bottlenecked at the point of transformation of the bacteria. However the yield could be improved by pooling multiple ligations (for this library we pooled three ligation reactions). We now acknowledge this limitation in the results section (page 13):

“To maximize diversity of the plasmid library the NNK mutagenesis saturation strategy was applied which theoretically covers all 20 amino acids at each of the 9 sites yielding a potential degeneracy at the amino acid level of 5.12×10^9 . It is clear that only a fraction of the combinations theoretically generated was represented in the recovered library and the efficiency could be improved with further optimization of the cloning process”.

“Since one candidate variant that had been selected (definition of the authors: selected more than once) showed a stop codon mutation, the question arises how the authors ensured that mutations presented at the capsid are encoded by the viral genome delivered by the respective mutant.”

As reviewer 2 points out, the escape of a ‘stop’-codon-containing capsid sequence among the sequenced colonies (AAV2BP5) would suggest that this genome was carry-over within a viable capsid. During the encapsidation of the viral components within the cell, multiple genomes are present within the HEK293 cell and it is not guaranteed that the genome encoding a capsid will be the same as the genome packaged within the encoded capsid. To minimize the number of such ‘carry-over’ sequences, the plasmid/(packaging-cell) ratio may be reduced as described by Maheshri et al, (Nat. Biotechnol. 24, 198–204). This latter approach is outlined in the methods section (page 13). Nonetheless it remains impossible to control the randomness of transfected plasmid uptake by the cell and of the encapsidation of the plasmids by the virus. It is thus only possible to verify whether a given sequence truly encodes the bipolar-cell-penetrating viral capsid by testing each cloned sequence individually.

In the revised manuscript we discuss the presence of the ‘stop’ escape (page 6):

“One variant (variant 5) carried a stop codon and was likely to have been carried through the screen when packaged in another viable capsid. During viral production in the packaging HEK293 cell, the stop-codon-containing sequence may have been taken up by a capsid encoded by another AAV genome within the cell. By using a low plasmid/(packaging-cell) ratio we hope to minimize the number of alternative AAV genomes within the cell. Nonetheless, due to the random nature of the transfection it is not guaranteed that the genome encoding a capsid will be the same as the genome the capsid then encapsidates. This most likely accounts for the presence of AAV8/BP5. We have confirmed that this serotype is non-viable in luminescence assay (Figure 1F)”.

However, we should point out that this stop-containing sequence was only found once in the sequences (see Table 2) and was included to make full disclosure. Thus as our criteria that variants were selected ‘because they appeared more than once’ was not met by this sequence we also propose its removal at the discretion of the editor and reviewer.

“how many particles within the initial library showed a wild-type sequence?”

We did not sequence any clones from the initial round of selection and used the DNA directly for the second round of library production. However the initial library was cloned in such a way as to avoid any bleed-through of WT capsid – we now highlight this in the methods section (page 13).

“What is the size of the viral genome packaged into the viral library? From the plasmid map it seems that the viral genome is oversized”

The plasmid map shown in Figure S5 is used only for subcloning of the library and was not directly used to make AAV. To make this ‘pre-library’ plasmid we added a stuffer element and a kanamycin-containing cassette between the *sbfl* and *mlu1* sites of the AAV8 capsid gene. We did this to minimize the risk of having single-cut plasmid backbone contaminating the gel fragment isolated following restriction digest with *sbfl* and *mlu1* (as otherwise the distinct sizes of any linearized backbone and double-cut backbone would be difficult to resolve on a gel). Even a single copy of full backbone would provide WT capsid that would contaminate and quickly saturate the library. By removing a larger 3.2kb kanamycin-containing element there would be no risk of carrying over partially-digested WT plasmid from the gel. As further control the colonies could be screened for no-growth on agar-kanamycin plates. The 370bp library fragment carrying the 9 NNK residues is cloned in place. Hence the final library-plasmid contains 4.27kb between the ITRs and is not too large to generate AAV. We have added this plasmid figure and legend (S5b) and added clarification to the methods section (page 13).

“What is the diversity of the library and which titer could be reached”

The diversity of the plasmid library is estimated at 2.6×10^5 cfu/ml. The functional titer we present for the viral library of 1×10^5 IU/ μ l is estimated from counting fluorescent spots on a plate of HEK293 cells transduced with serial dilutions of the AAV library. This is now described in the methods and results section (pages 5 and 13). We wished to use the same type of assay for estimation of viral library titer as for the estimation of library plasmid diversity; therefore the titer assay for the library is distinct from the qPCR and luminescence-based method used to establish titre for the individual AAVs.

“Since multiple virions can infect one cell, why do the authors speculate that mutant and wild type AAV will compete with each other for cell transduction”

We wished to test how the library compared to WT virus. As an alternative approach to pursuing multiple consecutive cycles of *in vivo* screening and library production we opted to directly identify which variants would emerge against a saturating background of WT capsid.

The strategy revealed interesting results immediately with only WT virus and no mutant viruses identified in the sequences from non-bipolar cells and 14 mutant sequences identified in the bipolar cells. This suggests that the saturating level of viral transduction, which could be achieved by WT virus in most cell types, was not attained in the bipolar cell. It is possible that an effective capsid sequence emerged independent of our selection rationale. However, our emphasis is on validation of AAV2/8BP2 as a bipolar-virus rather than on the process used to screen for this. We add a comment on the screening process in the discussion:

“Many variations on AAV libraries have been done to date and the success of these libraries relies heavily on the screening process used. We consider the *in vivo* screen using the *mglur6-gfp* mouse key to the isolation of this virus. Furthermore, the preferential uptake of WT AAV2/8 by non-bipolar cells validated a selection strategy whereby we could isolate variants against a saturating background of WT virus”.

“If ON-bipolar cells are poor target cells for AAV8 as stated on page 6, why is the wild type sequence the most prominent sequences that was selected (Table 2)”

This can be explained when we consider the vastly reduced titer of the variants compared to the wild-type: the wild-type titer was at 1×10^{12} gc/ml. Given nucleotide bias at each codon position and the different number of codons that code for each amino acid it is difficult to estimate the degeneracy of our library at the nucleotide level. However we can assume each individual variant to be present at far lower individual titers compared to WT.

We have now added this in the results section (page 6).

“It is notable that the wild-type capsid sequence was considerably more abundant (30 wild-type sequences versus 14 mutant sequences, Table 2). This was expected, due to the very dilute titer for each individual variant compared to the WT viral titre of 1×10^{12} gc/ml. Even taking into account the potential loss of diversity during viral packaging the emergence of any mutant sequence against a saturating WT background is significant.”

“How is it possible that BP5 could be packaged....(figure 1E and 1F)”

It is highly unlikely that BP5 could be packaged. This was reflected in the titration on HEK293 cells of figure 1E and 1F. Figure 1E showed vector yield 3 days post-triple-transfection and BP5 and BP6 emerged with lower values compared to the other variants. Originally we presented the data of figure 1E ‘GC/cell’ based on microtiter assay. However the range of titre is narrower for the microtiter assay; to avoid confusion we now present the data as GC/ml looking at expression from a total of 5×10^4 HEK cells. The figure legend and methods have been adjusted accordingly. We also obtained a 10-fold lower infective titre upon luminescence readings of cells transduced with BP5 or BP6 compared with BP2 (figure 1F) and this matches background. We thus considered BP5 as a negative control and this prompted us to discard BP6 from further screening.

“On the same line, how was it possible to select this sequence”

It is assumed that the BP5 sequence was carried over in capsids that were encoded by other AAV genomes in the HEK cell. The random residues NNK would prevent the encoding of two stop residues but would still allow for TAG. This residue may arise 1.56% of the time (1/64) such that across 9 residues its frequency will be 14%.

We have now discussed this on page 6

“One variant (variant 5) carried a stop codon and was likely to have been carried through the screen when packaged in another viable capsid. Its persistence in the screen may be accounted for when you consider that the TAG residue may arise 1.56% of the time (1/64) such that across 9 residues its frequency will be 14%. It served as a negative control for titration”.

“From the text it is not clear whether comparable particle numbers are injected...(Figure 2 and S3)”

We have clarified this in page 7.

“For this initial analysis, the normal recombinant AAV2/8 virus was injected at a titer of 1×10^{13} gc/ml and the variant viruses injected at a titer of 2×10^{12} gc/ml”.

“Did the author observe any cell transduction outside the retina”

We did not perform a biodistribution analysis as we had a very specific goal for this virus, that of vision restoration.

“The authors state that they used a low plasmid concentration to avoid packaging of multiple genomes into each capsid. It is not possible to package multiple full-length genomes into a single AAV capsid. Thus, this statement needs to be corrected.”

I am grateful to this reviewer for identifying this erroneous statement. It relates to a critical technical detail. We have now amended this statement as follows:

“To minimize the transfection of multiple rep/cap genes into each cell (which would risk encapsidation of a non-‘self-coding’ genome), a low plasmid concentration was used for the triple transfection”

3) Minor points:

Reviewer 1 requested clarification on the nature of the enhancer element used with the 4xMG6 promoter. This is addressed in the results section:

“Ueda *et al* (1997) generated transgenic mice using the 5' flanking mouse *Grm6* sequence fused to a reporter gene. This 10-kb region was capable of directing the cell type-specific and developmentally regulated expression of the *Grm6* gene. Kim *et al* refined the sequence to a 200-bp critical enhancer region (−8126 to −7927 relative to the first nucleotide of GenBank accession number BC021919, NIH) that could be used to drive reporter gene expression specifically in ON-bipolar cells (Kim *et al.*, 2008; Lagali *et al.*, 2008)”.

An effort has been made to eliminate any unreferenced statements. In the methods section we have described the source of the HEK293 cells used to make the AAV.

“The HEK293 cells express the helper E1A/E1b gene (Stratagene, catalogue number 240073)”.

As suggested by reviewer 1 we have had the paper reviewed by an undergraduate to help us identify those areas that were unclear. We have added a number of references and have endeavoured to improve the overall clarity by defining abbreviations.

As reviewer 2 suggests we have added further serotypes and their references into our introduction to 3D models and have added references for targeted mutagenesis of other serotypes on page 4. We have replaced ‘yellow box’ with ‘red box’ in the figure legend for figure 3H and the label of Figure S6 has been added.

Thank you for giving us the opportunity to revise this paper. We hope to have adequately addressed your concerns and those of the reviewers.

2nd Editorial Decision

23 June 2014

Thank you for the submission of your revised manuscript to EMBO Molecular Medicine. We have now received the enclosed reports from the referee who was asked to re-assess it. As you will see this referee is now globally supportive and I am pleased to inform you that we will be able to accept your manuscript pending the following final amendments:

1) please address all the remaining issues highlighted by the reviewer by correcting the text when suggested and adding explanations when needed.

***** Reviewer's comments *****

Referee #2 (Remarks):

The authors submitted a revised manuscript and point-by-point reply to my concerns, questions and comments.

However, a few issues remained.

Figure 1 is confusion since infectivity of a small-scale production and genomic titers of large-scale productions are show. Based on the main text, a large-scale production has only been performed for BP2 and was done following in vivo comparison of the mutants.

Thus, when comparing results of the small-scale production AAV2/BP4, AAV2/BP6 (and AAV2/BP3) showed background level (Fig. 1E, S3), which based on their own statement on BP5 was judged as non-viable. Only when judging the large-scale preparation, BP3, BP4 and BP6 showed a higher titer compared to BP5. The authors should more precisely explain based on which criteria the mutants (other than BP5) were selected or excluded, and should mentioned that both, a large- and a small-scale preparation was performed prior to taking this decision. Were the large- or small-scale preparations used for experiments depicted in Fig. 2 and S4?

There is a break in logic. Since AAV2/8-BP5 is a stop codon mutant and the authors state that it is therefore used as negative control (page 7) there is no reason to state that this mutant was not further tested since it yielded a low titer (page 8). On the same line, the last sentence of the paragraph >Strategy to modify AAV tropism < already announces that only four variants had been selected for further investigations based on titration results. It is therefore confusing to state in the following paragraph that two mutants were excluded because of the low yield. This should be revised.

The sentence >This virus was confirmed to transduce HEK293 cells in vitro in a similar way to the parental AAV2/8, suggesting that the modified epitope was not having a significant negative impact on viral packaging (Figure 2C)< (page 9) has to be revised to >This virus was confirmed to transduce HEK293 cells in vitro in a similar way to the parental AAV2/8, suggesting that the modified epitope was not having a significant negative impact on the natural tropism (Figure 2C)<. If the same number of genome containing particles per cell is applied, transduction will reveal the amount of vectors that successfully deliver vector genomes, however, it is not a measure on how efficient vector genomes have become packaged during the production procedure.

Page 9: Did the authors match the infectious or the genomic titers?

The authors demonstrate successful transduction of human bipolar retinal cells with AAV2/8-BP2. How efficient is AAV2/8 in transducing these cells? Does AAV2/8 lack the ability to efficiently transduce both, mouse and human bipolar retinal cells?

The infectivity of the AAV2/8 in comparison to the mutant on a non-target cell is only of relevance if due to the 9mer insertion, virus tropism of the mutant is not affected or has been expanded. If the tropism has been changed, infectivity is maybe under- or overestimated. If the infectivity is underestimated, mutants that did not show infectivity on HEK293 are maybe wrongly excluded. Thus, in all experiments equal physical (genomic) particle numbers should be applied in those experiments that aim to compare AAV2/8 and mutants side-by-side. Furthermore, results presented for example in Fig. 1 should be discussed respectively. The same holds true for the authors attempt to determine infectivity of the library.

Bacteria are quite fast dividing organism. Can the authors exclude that multiple divisions have taken place prior to seeding the library starter culture aiming to estimate the library diversity?

Minor

Viral mutants were isolated three weeks post injection. This strategy was successful. However, since it is an unusual long time period compared to previous selection strategies, the authors may include a comment explaining why they decided to do it in this way.

Thank you for accepting our manuscript, "Efficient transduction and optogenetic stimulation of retinal bipolar cells by a synthetic adeno-associated virus capsid and promoter".

In the final version we have addressed the reviewers remaining issues:

- 1) The scale of viral production for all the variants has been clarified and we have rephrased our description of exclusion criteria for the variants on page 8. (*"The variant sequences were processed for further analysis, and four variants were further selected based on the titration data from small-scale (Figures 1E and S3) and large-scale preps (figure 1F) as well as structural predictions of the targeted region anticipated to potentially affect receptor/ligand interactions (Figure S2)"*).
- 2) As proposed we have corrected the sentence *"This virus was confirmed to transduce HEK293 cells in vitro in a similar way to the parental AAV2/8, suggesting that the modified epitope was not having a significant negative impact on the natural tropism"* (page 9)
- 3) We clarified that genomic titres were matched (page 9).
- 4) We have added a reference to show the lack of AAV8 transduction holds true for human retinal explants (Busskamp et al, 2010, page 15).
- 5) We have described how viral mutants were isolated three weeks p.i. in order to allow time for onset of maximal transgene expression and we have provided an additional reference (page 6).

We are very happy to have this article published in EMBO Molecular Medicine. We note that there is intense competition to achieve vision restoration via bipolar-cell mediated optogenetics and hope that the paper might become available ahead of time.